



# Quantifying organic matter and functional groups in particulate matter filter samples from the southeastern United States – Part 2: Spatiotemporal Trends

Alexandra J. Boris[1], Satoshi Takahama[2], Andrew T. Weakley[1], Bruno M. Debus[1], Stephanie L. Shaw[3], Eric S. Edgerton[4], Taekyu Joo[5], Nga L. Ng[5,6,7], Ann M. Dillner[1]

[1]Air Quality Research Center, University of California Davis, Davis, California USA
[2]Ecole Polytechnique Federale de Lausanne, Lausanne, Switzerland
[3]Electric Power Research Institute, Palo Alto, CA, 94304, USA
[4]Atmospheric Research & Analysis, Inc., Cary, NC 27513, USA
[5]School of Earth and Atmospheric Sciences, Georgia Institute of Technology, Atlanta, GA 30332, USA
[6]School of Chemical and Bimolecular Engineering, Georgia Institute of Technology, Atlanta, GA 30332, USA
[7]School of Civil and Environmental Engineering, Georgia Institute of Technology, Atlanta, GA 30332, USA

*Correspondence to*: Ann M. Dillner (amdillner@ucdavis.edu)

**Abstract.** Organic species within atmospheric particles vary widely in molecular structure. The variety of molecules that comprise the aerosol make it rich in information about its sources and chemical lifecycle but also make particulate organic matter (OM) difficult to characterize and quantify. In Part 1 of this pair of papers, we described a direct method for measuring the composition and concentration of OM in aerosol samples that is compatible with routine monitoring of air quality. This method uses Fourier Transform Infrared (FT-IR) spectrometry of filter-based aerosol samples to quantify bonds, or functional groups, that represent the majority of organic composition; summation of these functional groups gives OM. In this paper, functional group composition and OM concentrations are directly measured in eight years of aerosol samples collected at two rural and two urban sites in the Southeastern Aerosol Research and Characterization (SEARCH) network. FT-IR spectrometry with a multivariate calibration is used to quantify the concentrations of aliphatic C-H (aCH), carboxylic acid (COOH), oxalate (oxOCO; representing carboxylates), non-acid and non-oxalate carbonyl (naCO), and alcohol O-H (aCOH) in approximately 3500 filter samples collected every third day from 2009 through 2016. In addition, measurements are made on samples from all days in 2016.

A decline in the total OM is observed from 2011 to 2016 that is caused by decreases in the more oxygenated functional groups (carboxylic acid and oxalate) and is attributed to anthropogenic $SO_2$ and/or volatile organic compound (VOC) emissions reductions. The trend in OM composition is consistent with those observed using more time- and labor-intensive analytical techniques. Concurrently, the fractional contributions of aCOH and naCO to OM increased, which might be linked to monoterpene-derived secondary OM, with possible influences from decreasing $NO_x$ and/or increasing $O_3$ concentrations. In addition, this work demonstrates that OM to organic carbon (OM/OC) ratios in the Southeast U.S. (SE U.S.) did not appreciably change over the study time period, as a result of these competing functional group contributions to OM. Monthly observations support the sources suggested by these overall trends, including strong biogenic and photo-oxidation influences,





while daily samples from 2016 further elucidate the consistent impact of meteorology and biomass burning events on shorter term OM variability, including prescribed burning in the winter/spring and wildfires in the autumn. These shorter-term and spatial observations thus reinforce the results of the broader dataset and serve to evaluate the applicability of FT-IR spectrometry measurement to trends analysis on various timescales relevant to routine monitoring of aerosol composition.

## 1 Introduction

### 1.1 A functional group perspective on atmospheric aerosol organic matter

The composition of atmospheric aerosol organic matter (OM) is rich in information about the sources and processes leading to particle phase pollution (Rogge et al., 1993). However, OM is not directly quantified in monitoring networks and many intensive field campaigns. While organic carbon (OC) concentrations are often measured in such cases, OC captures only the variability in carbon atom concentrations, and the contributions of oxygen and hydrogen atoms are assumed based on a single OM/OC ratio. However, this OM/OC ratio varies substantially in atmospherically relevant molecules (Turpin and Lim, 2001); within the ambient atmosphere, the ratios are also estimated to vary strongly between seasons and sites (Simon et al., 2011). Thus, a direct method for regularly and directly quantifying OM in aerosol samples would improve the accuracy of measured OM concentrations and trends.

Much of the work on OM mass has focused on speciation of individual compounds, so that only a portion of OM mass is measured (using chromatography and mass spectrometry, MS, techniques, for example, by Gao et al., 2006), or modelling approaches based on concentrations of speciated chemical tracers (Budisulistiorini et al., 2015; Kleindienst et al., 2010). Comprehensive OM quantification can be afforded by other techniques that are not readily applicable in routine measurements because of expense and labor requirements (for example, proton transfer reaction MS and aerosol MS; Ditto et al., 2018; Salvador et al., 2016; Schum et al., 2018; Xu et al., 2015a; Zhang et al., 2018). For routine monitoring and many field campaigns, measuring bulk chemical properties of OM could be an appropriate middle ground between estimating OM from OC measurements and speciating a small fraction of the OM by mass. Organic functional groups (FGs) have been explored as such a middle ground, quantifying the composition of OM from distinct sources (Coury and Dillner, 2009; Russell et al., 2011; Takahama et al., 2011) as well as measuring the age, or degree of oxidation, of sampled OM (Russell, 2003; Ruthenburg et al., 2014; Turpin and Lim, 2001).

The conceptualization of OM in terms of FGs differs from that of molecular composition, as used with chromatography methods, and is more akin to direct spectral results of hard ionization mass spectrometry. Multiple FGs are present in every molecule, so the contributions of FGs to each sample are indicative of OM composition; a greater fraction of oxygenated FG content can indicate more aged material (Chhabra et al., 2011; Moretti et al., 2008). By studying the FG composition of OM within atmospheric aerosol samples, patterns emerge in the bulk OM composition, from which sources and formation processes can be revealed (Corrigan et al., 2013; Liu et al., 2018; Russell et al., 2009; Takahama et al., 2011).



Fourier transform infrared (FT-IR) spectrometry is used to quantify FG concentrations in aerosol filter samples (Coury and Dillner, 2008; Maria et al., 2002; Reggente et al., 2018; Russell et al., 2009). The analysis can be performed on polytetrafluoroethylene (PTFE) filters without damaging the sample; this enables the filters to be archived and/or used for other analyses, making FT-IR spectrometry suitable for monitoring networks and field campaigns that collect PTFE filter

5      samples.

### 1.2 Unique and changing atmospheric chemistry and climate of the southeast U.S.

The hot, humid climate in the southeast (SE) U.S. creates a unique environment for mixing and reaction of anthropogenic and biogenic aerosol chemical components, including within liquid water, as well as the generation of oxidants (Hansen et al., 2003). Stagnation events occur both in the summer and winter, causing build-up of atmospheric pollutants (Hansen et al.,

10     2003). In addition, aerosol composition and emissions sources of the SE U.S. are distinct. The Southeastern Aerosol Research and Characterization (SEARCH) network was designed to study atmospheric chemistry in the SE U.S., at urban/rural pairs from 1998 through 2016 (Figure 1; Blanchard et al., 2013a; Edgerton et al., 2005, 2006; Hansen et al., 2003). The configuration of sites varied over the years, but the longest-operated sites include the urban/rural pairs in Alabama and Georgia (Figure 1).

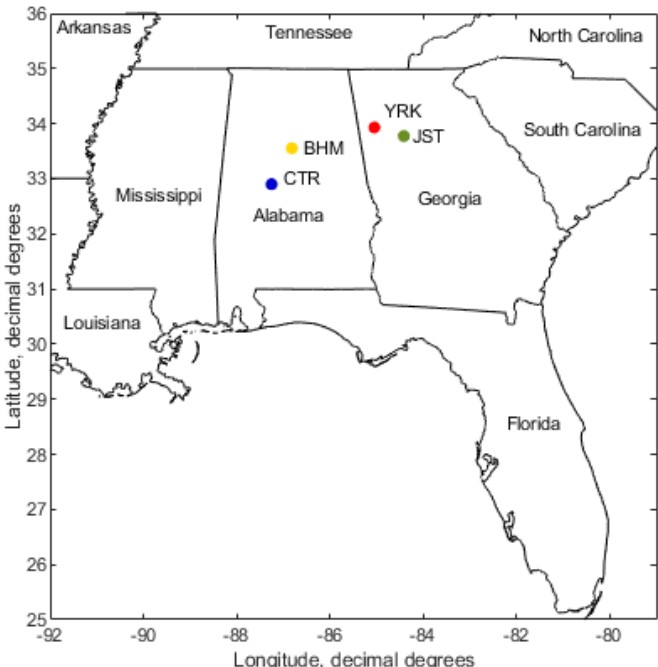

**Figure 1. Map of SEARCH network showing urban and rural sites: YRK = Yorkville, GA (rural); JST = Jefferson St., Atlanta, GA (urban); CTR = Centreville, AL (rural); BHM = Birmingham, AL (urban). Map generated using the borders function created for Matlab (Greene et al., 2019).**



The Yorkville, Georgia (YRK) site was located ~55 km from a major urban area (Atlanta), and was surrounded by pastures and nearby forest (~100–300 m from samplers). Some agricultural emissions might be expected at YRK, while the Alabama rural site, Centreville (CTR), was surrounded by forest (Edgerton et al., 2005). The CTR site was ~85 km from its paired urban area (Birmingham or BHM), with only a few residences in the area; the CTR site therefore represents a more rural

location than YRK. Analysis methods were advanced and comprehensive, including real-time gas phase measurements, light and mass-based measurements of total particles, a variety of particle-phase composition measurements (trace elements, inorganic salts, OC, and elemental carbon), and supporting meteorological variables (Hansen et al., 2003).

A partial characterization of SE U.S. OM in previous years can be drawn from studies within the SEARCH network data and the many field campaigns carried out in the region (Carlton et al., 2018; Edgerton et al., 2005; Kim et al., 2015; Liu et al.,

2018; Mao et al., 2018; U.S. Environmental Protection Agency, 2011; Xu et al., 2015a). The dominant contribution of secondary material to OM and a regional homogeneity of primary OM sources has been demonstrated (Chen et al., 2012). There are also unique emissions sources and secondary pollutants in the SE U.S.; in particular, prevalent tree species emit high isoprene and monoterpene loadings relative to other regions (Guenther et al., 1994, 2012) and contribute substantially to the OM (Xu et al., 2015b; Zhang et al., 2018). Prescribed burns, wildfires, and residential wood burning smoke sources were

demonstrated to be strong contributors of primary aerosol loadings at SEARCH sites (Blanchard et al., 2013a; Chen et al., 2012). Fossil fuel combustion may contribute substantially to OM in the SE U.S., especially at urban sites: ~50 % of primary and secondary OC at urban BHM was estimated to be from fossil sources, but at rural CTR, >80 % of primary and all of secondary OC was from modern sources such as trees (in the early 2000s; Blanchard et al., 2008). However, since the late 1980s, controls on anthropogenic emissions have resulted in VOC, $NO_x$, OC, $SO_2$ $SO_4^{2-}$, and other species concentration

declines (Blanchard et al., 2014, 2016; Hand et al., 2012a; Hidy et al., 2014). More recent work by Chen et al. (2020; in 2016 at YRK) confirmed that secondary OM from biogenic carbon sources continues to be important in the SE U.S.

Several SE U.S. studies have highlighted decreasing OM concentrations due to the interactions between anthropogenic and biogenic pollutants (Goldstein et al., 2009). In particular, declining $SO_2$ emissions are likely to be driving declines in biogenic OM concentrations (Marais et al., 2017; Xu et al., 2015b). The result of declining $NO_x$ concentrations on OM in the

SE U.S. has also varied among studies (Pye et al., 2019; Travis et al., 2016; Wolff et al., 2013; Zhang et al., 2018), with trade-offs between the availabilities of major oxidants and volatilities of products at varying $NO_x$ concentrations. These changes in chemical climatology make the SE U.S. a unique location for studying a method to characterize OM composition, as well as examining the effects of declining anthropogenic emissions on atmospheric chemistry.

### 1.3 Research Statement

We herein present the application of FT-IR spectrometry to eight years of routinely collected SEARCH network PTFE ambient aerosol filter samples to examine trends in OM concentrations, OM/OC and their FG composition from 2009 to 2016. FG calibration models described by Boris et al. (2019; i.e., part I of this study) were applied to SEARCH spectra in the current study to accomplish the following goals: (1) examine the yearly, seasonal, and daily trends in OM and its FG





composition; (2) characterize the trends in directly measured OM/OC over the eight years at all sites; and (3) evaluate the applicability of the FT-IR spectrometry method for routine OM characterization. The comparison of the results to other FG studies, SEARCH network data, as well as data from other studies of SE U.S. atmospheric chemistry provided context for these measurements and suggested explanations for the observed trends.

## 5   2 Methods

SEARCH network filter samples of atmospheric particulate matter with ≤2.5 μm aerodynamic diameter ($PM_{2.5}$) were obtained from Atmospheric Research & Analysis, Inc. (Cary, NC). Samples from every third day in 2009–2015 and every day in 2016, from five sites in the SE U.S., were analysed during this study (Sect. 2.1) along with additional samples collected at one SEARCH site in 2018 (Sect. 2.2). Additional datasets are used to support findings (Sect. 2.3). The samples

were analyzed using FT-IR absorption spectrometry (Sect. 2.4), and concentrations of organic FGs were calculated using multivariate calibration models (Boris et al., 2019). Organic matter (OM) concentrations were calculated for each sample as a linear combination of the FG concentrations measured (Sect. 2.7). Spatial and temporal trends were explored in the resulting OM concentrations, OM/OC, and contributing FG quantities (Sect. 2.8).

### 2.1 SEARCH aerosol samples and network data

Filter samples of ambient aerosol were collected in the SEARCH network on MTL 47 mm PTFE filters with 2 μm pore size (Measurement Technology Laboratories, https://mtlcorp.com/filters) at 16.7 liters per minute using Partisol Plus 2025 samplers (Rupprecht & Patashnick/Fisher Scientific, http://www.thermofisher.com/). SEARCH samples from 2009–2016 were used in the present study (samples were also collected between 1999 to mid–2008 but were not refrigerated during storage so may have compromised organic composition). Although the paired methods paper reports only analysing samples

from one in three months per year, to better elucidate trends, samples from every one in three days in all months 2009–2015 (a sampling schedule matching that for thermal optical reflectance (TOR) analysis for OC measurements) were used in this study; samples for all days were included during 2016. The FTIR analysis of the additional months was performed at least one year after the initial set of samples were analysed. During this additional analysis, samples collected in 2011, 2012, and 2014 at Centreville, AL were erroneously kept in room temperature storage, rather than kept cold, between ~August and

November 2018. All told, at each site, ~100 (68–123) samples were analyzed by FT-IR spectrometry per year from 2009 to 2015. In total, 3508 samples from 2009–2016 with and 1267 samples from 2016 were analyzed. The SEARCH network ended sampling in 2016 at BHM on October 18th. Three sites in addition to those in this work were closed before 2016. Multiple chemical and physical parameters were measured by the SEARCH network (discussed in detail by Hansen et al., 2003; filter-based measurements by Edgerton et al., 2005, and continuous gas- and particle-phase measurements by Edgerton

et al., 2006). Gravimetric analysis of $PM_{2.5}$ mass and x-ray fluorescence of trace metal concentrations were performed using the same Teflon filters analyzed by FT-IR spectrometry. TOR analysis for OC and elemental carbon (EC) concentrations





was performed using 37 mm quartz filters (and OC concentrations were blank corrected by subtracting network-wide annual mean field blank OC concentrations). Ions including $SO_4^{2-}$, $NO_3^-$, and $NH_4^+$ were measured using ion chromatography from 47 mm PTFE filters; negative artefacts of $NO_3^-$ and $NH_4^+$ were quantified from 47 mm Nylon and cellulose filters, respectively.

**2.2 Additional dataset: January and February 2018 at Atlanta Site**

Daily samples were collected at JST from January 15 to 21, January 29 to February 4, and February 12 to February 25, 2018. Samples were collected in a similar manner to those collected in the SEARCH network including sampling for 24 hours with a Partisol 2025 particle sampler loaded with MTL 47 mm diameter PTFE filters (as described above).

**2.3 Supporting datasets**

In addition to chemical data, meteorological variables were gathered at the SEARCH network sites. However, precipitation data were not available. Precipitation observations were therefore retrieved from the National Oceanographic and Atmospheric Administration website (https://www.climate.gov/maps-data/dataset/past-weather-zip-code-data-table). Satellite imagery was used to determine whether smoke plumes from wildfires and/or prescribed burns were observed on each sample date; these images were retrieved from National Aeronautics and Space Administration WorldView (https://worldview.earthdata.nasa.gov/) using the Aqua MODIS Corrected Reflectance base layer.

**2.4 FT-IR spectrometry analyses**

All filter samples and standards were analyzed using a Bruker Tensor II FT-IR spectrometer (Bruker Optics, Inc.; http://www.bruker.com/) operated in transmission mode. The spectrometer was equipped with a liquid nitrogen cooled mercury cadmium telluride detector. Filters were placed into a sample chamber within the spectrometer that was custom-built (Debus et al., 2018); the chamber was flushed continuously with air scrubbed of $H_2O$ and $CO_2$ (model VCDA air purge system, Puregas, LLC, http://www.puregas.com/; <10% humidity). The FT-IR spectrometry analyses have been described in more detail elsewhere (Takahama et al., 2013). Spectra used in the present work include the wavenumber range 4000 to 1500 cm$^{-1}$; the range of 1500 to 400 cm$^{-1}$ was excluded due to strong absorption by the PTFE filter substrate (Mayo et al., 2003) and highly specific absorption for each chemical. Samples were re-analyzed after 1–2 years to demonstrate the stability of the samples and resulting spectra/concentrations; handling/analysis and long-term storage did not substantially affect the spectra, but a small bias (-5 to -10 % per year after the first two years) was observed, likely due to storage and/or transport of samples (Boris et al., 2019). Outliers were detected and removed to minimize the influence of questionable data on trends (Boris et al., 2019).


## 2.5 Outlier detection and handling

Outlier samples and field blanks, such as ripped filters or swapped samples, were identified using the following criteria. First, when available, if the observed flow rate, duration, or total volume was not 100±5 % of the expected value (e.g., <23.75 of 24.0 expected hours), the sample was excluded from the dataset; samples with three or more null flow observations were assumed to be inaccurately sampled and were also excluded. In addition, field blanks with clear collected aerosol material were removed. Samples with spectra that were obviously collected incorrectly were also excluded (e.g., no filter was loaded into the chamber and the spectrum contained no expected absorption features). Samples with the following characteristics were also flagged and removed if the sample was definitely collected incorrectly (e.g., an anomalous FT-IR spectrum and poor predictions): (1) a spectrum corresponding to a high TOR OC concentration, but low infrared absorption; or (2) a high error in prediction after calibration. Overall, approximately 5 % (187/3661) of the ambient 2009–2016 1:3 day samples were removed from the dataset; approximately 5 % (70/1337) of the ambient 2016 daily samples were removed. Samples missing TOR OC concentrations or other network measurements were still included in the results.

## 2.6 Functional group calibration models

FG concentrations were quantified in the ambient samples using multivariate calibration models (methods are described in detail by Boris et al. (2019). The calibration models were constructed using laboratory standards of 17 organic compounds and 3 inorganic interferents, as discussed in our methods paper. Five FGs were quantified and reported using separate FT-IR spectrometry/partial least squares (PLS) calibration models: aliphatic C-H (aCH), carboxylic acid (COOH), carboxylate represented by oxalate (oxOCO), non-acid non-oxalate carbonyl (naCO), and alcohol O-H (aCOH). One model was constructed for each FG with the exception of naCO, for which a linear regression between carboxylic acids and total carbonyls was used to determine the concentration (Boris et al., 2019; Takahama et al., 2013). The spectra and masses were regressed using a multivariate PLS algorithm (Wold and Sjostrom, 2001) to obtain a set of regression coefficients describing each FG (Reggente et al., 2018; Ruthenburg et al., 2014). Sample concentrations of each FG were then determined by applying the regression coefficients to each sample FT-IR spectrum. Because there were no directly comparable FG concentrations for these samples, models were evaluated using a variety of metrics, such as TOR OC and residual OM concentrations (the latter is calculated by subtracting a weighted sum of major inorganic ionic and elemental aerosol components and elemental carbon from $PM_{2.5}$ mass for each sample, as described by Hand et al., 2012b).

## 2.7 Calculation of OC and OM concentrations

Concentrations of OC and OM were calculated as linear combinations of the measured FG concentrations (Boris et al., 2019). OC concentrations were calculated as the summation of carbon masses contributed by each quantified FG, assuming the following ratios of carbon atoms per bond: aCH=0.5C (i.e., on average, C-H bonds were assumed to be –$CH_2$-), COOH=1C, oxOCO=1C, naCO=1C, and aCOH=0.5. The same ratios were used by Russell et al. (Russell, 2003) and Boris





et al. (2019), and were supported by modelling work (Takahama and Ruggeri, 2017). In contrast, Ruthenburg et al, 2014 assumed that aCOH contributed no carbon (assumed to be primary or secondary). OM concentrations were similarly calculated as the sum of each of the FG contributions to O, H, and carbon mass. The same assumed ratios for carbon contributions were applied, plus all associated O and H atoms. OM/OC ratios were calculated from these OM and OC concentrations; contributions of each functional group to OM/OC were calculated as the contributions of O and H mass to OC, with carbon mass contribution to the OC to one (as in previous representations; e.g., Takahama and Ruggeri, 2017).

### 2.7.1 Method detection limits and sample uncertainty

The method detection limit (MDL) for each FG was estimated as the difference between the 95th and 50th percentiles of FG concentrations (UC Davis Air Quality Research Center, 2017) measured from 20 laboratory blanks and 616 field blanks. The MDLs of the calculated OM and OC concentrations were estimated using the same blanks, as the root of the sum of squares of the FG MDLs (Boris et al., 2019). Samples with FG or OM concentrations less than the value of the respective MDLs were censored as described by Boris et al., 2019. For all FGs, >85% of sample concentrations were above the MDL (97% oxOCO, 95% aCH, 86% aCOH, 92% COOH, and 89% naCO).

The precision of the method (including aerosol sampling, differences in filter substrates, handling, and FT-IR spectrometry analysis) was evaluated by comparing measurements from two collocated sampling sites within the SEARCH network. This "sampling uncertainty" was calculated (Hyslop and White, 2008, 2009) and FG concentrations from samples collected at the JST site and its collocated site, cJST, during 2009–2011, October 2015, and 2016.

### 2.8 Annual, monthly and daily concentrations and trend values

Median values were used to report annual and monthly functional group and OM concentrations, to minimize the impact of extreme values. Theil–Sen rank-invariant regression (Sen, 1968; Theil, 1950) was used to determine the slopes of multi-year trends in measured variables. The slopes were expressed as change in concentration per year ($\mu$g m$^{-3}$ yr$^{-1}$) or change in fractional FG contribution to OM per year (%OM yr$^{-1}$). Confidence intervals about each slope were calculated as 95th and 5th percentiles of bootstrapped slopes and intercepts ($n$=2500; method for confidence interval estimation was carried out following the percentiles method for Theil–Sen regressions described by Wilcox, 1998).

### 3 Results and discussion

Declining trends in aerosol OM concentrations were observed in the SE U.S. in urban and rural locations Table 1; Sect. 3.1). The OM composition also changed with distinct declines in contributions from COOH and oxOCO but enhancements in naCO and aCOH contributions (Table 2, Sect.s 3.2 and 3.3). Calculated OM/OC ratios were similar in magnitude to previously observed SE U.S. values, but multi-year trends differed from those estimated using TOR OC and other


measurements (Sect. 3.4). The seasonality of the OM composition, as well as changes in the seasonality over the 2009–2016 period (Sect. 3.5), supported mechanisms suggested for the longer-term trends. The daily 2016 dataset (Sect. 3.6) was used to demonstrate clear patterns in OM concentration build-up and washout, especially related to smoke, and differences in OM dynamics by season.

### 3.1 Trends of OM: Corroboration of chemical climatology change as a result of anthropogenic emission declines

Overall, decreasing concentrations of OM measured by FT-IR spectrometry ("FG OM") were observed between 2009 and 2016 in the SE U.S. ($-0.04 \pm 0.02$ µg m$^{-3}$ yr$^{-1}$, -8 % total). This change in OM concentration has a pattern similar to that of the concentrations of anthropogenic inorganic pollutant species, which also decreased in abundance over the study period, as shown in Figure 2. Declines in concentrations were observed for total PM$_{2.5}$, particle NO$_3^-$, SO$_4^{2-}$, elemental carbon (EC), and NH$_4^+$ in SEARCH data and other SE U.S. aerosol samples (Blanchard et al., 2013a; Feng et al., 2020; Malm et al., 2017; Marais et al., 2017).

OM concentrations increased during the 2008–2010 recovery from an economic recession (Feng et al., 2020; de Foy et al., 2016; Hand et al., 2019; Hidy et al., 2014), so multi-year trends discussed in further paragraphs will be expressed for the latter six year period of 2011 through 2016 (Table 1). The resulting slope in OM ($-0.08 \pm 0.07$ µg m$^{-3}$ yr$^{-1}$, or -18 % over the six year period) is similar in magnitude to those estimated over previous decades (e.g., -1.9 % yr$^{-1}$ OM 1998–2013 using SEARCH data, Marais et al., 2017, compared to -2.9 % yr$^{-1}$ OM 2011–2016 in the current study).





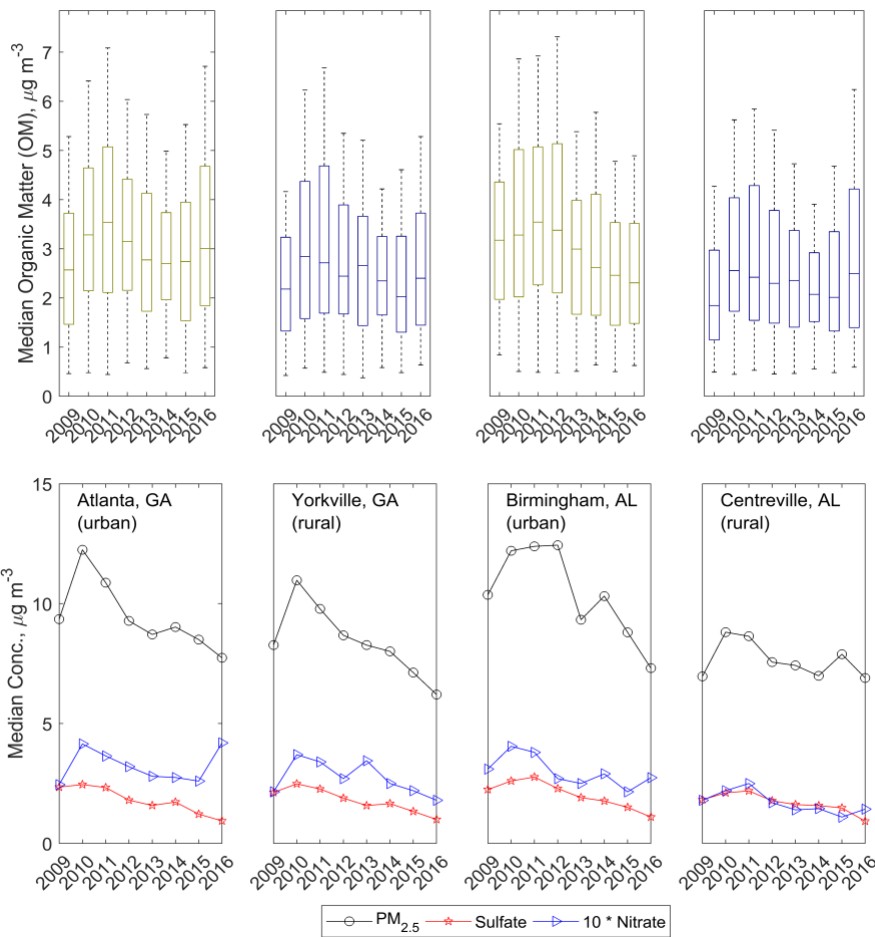

**Figure 2. Top: Annual median OM concentrations quantified via FT-IR spectrometry; colors indicate rural (dark blue) or urban (olive) site locale. Bottom: annual median total PM$_{2.5}$ concentration, particle SO$_4^{2-}$, and particle NO$_3^-$ concentrations as measured in the SEARCH network. Column-wise subpanels correspond to individual sites, as indicated in the bottom row.**

Particle-phase concentrations of non-polar organic species also decreased between 2006 and 2010 (Blanchard et al., 2014), as did non-methane VOC concentrations between 1999 and 2008 (Hidy et al., 2014). A slight increase in the contribution of OM to PM$_{2.5}$ concentrations was observed (Figure 2), which has been attributed especially to the strong decline in SO$_4^{2-}$ concentrations, and therefore enhanced contribution of OM to PM$_{2.5}$ (Marais et al., 2017). In contrast, mean 24 h mixing ratios of O$_3$ in the SEARCH network increased slightly over the studied time period (approximately +0.03 ppb yr$^{-1}$), with strong enhancements in springs and strong declines in summers (+0.79 ppb yr$^{-1}$ in springs over all sites; summers -1.17 ppb yr$^{-1}$, autumns +0.10 ppb yr$^{-1}$, and winters +0.49 ppb yr$^{-1}$). Coincident with O$_3$ declines, summertime OM concentration declines were strongest (-0.24±0.20 µg m$^{-3}$), supporting O$_3$ as an influential constituent in controlling OM concentrations (Table 1; Sect. 3.2).



**Table 1. Trends in variables measured using FT-IR spectrometry. Slopes were calculated using Theil–Sen regression and expressed between 2011 and 2016 (Sect. 2.8). For FG OM and FG OC, the season and site with the most extreme slopes are shown separately from "All Others". Confidence limits are bootstrapped at 95 % confidence. Slope values in bold are greater than the confidence limits. Note that units for OM/PM$_{2.5}$ and OM/OC values are yr$^{-1}$.**

| Variable | Sites | Seasons | Slope ± Confidence Limit ($\mu g\ m^{-3}\ yr^{-1}$) |
|---|---|---|---|
| FG OM | All | All | **-0.08** ± 0.07 |
| FG OM | BHM | All | **-0.19** ± 0.18 |
| FG OM | All Others | All | -0.08 to -0.02 ± 0.17 |
| FG OM | All | Summer | **-0.24** ± 0.20 |
| FG OM | All | All Others | -0.06 to -0.03 ± 0.16 |
| FG OC | All | All | -0.04 ± 0.04 |
| Nitrate | All | All | **-0.025** ± 0.008 |
| Sulfate | All | All | **-0.25** ± 0.05 |
| Ammonium | All | All | **-0.090** ± 0.018 |
| TOR EC | All | All | **-0.044** ± 0.014 |
| TOR OC | All | All | **-0.10** ± 0.07 |
| PM$_{2.5}$ | All | All | **-0.58** ± 0.21 |
| OM/PM$_{2.5}$ | All | All | **0.007** ± 0.006 yr$^{-1}$ |
| OM/OC | All | All | 0.002 ± 0.008 yr$^{-1}$ |

Changes in SE U.S. aerosol and OM concentrations have been modelled and discussed with respect to several alternative sources to direct anthropogenic emissions declines. The observed change in OM abundance was not due to the overall decreasing PM$_{2.5}$ concentrations: Si and K concentrations, for example, did not significantly decrease during the study period. Decreasing anthropogenic PM$_{2.5}$ concentrations are expected to be offset by increasing wildfire emissions globally (Mehta et al., 2018), and especially in the SE U.S. (Ford et al., 2018). However, overall OM and PM$_{2.5}$ concentrations in the

present study period have declined, demonstrating that any enhancement due to biomass burning emissions had not yet exceeded the impact of regulatory success in decreasing air pollution emissions. In agreement, smoke OM also did not appear to change appreciably between 2000 and 2011 (Blanchard et al., 2013a).

The overall decline in OM concentrations was observable for all four sites in the annual medians (Figure 2), with the largest at BHM and the smallest at the most rural of the sites studied, CTR (only the BHM site-specific OM trend was statistically

significant; Table 1). The OM was somewhat regional in character, as suggested in other SEARCH network research (Hidy et al., 2014): coefficients of determination ($R^2$) between urban and rural site OM concentrations were 0.69 (Georgia sites) and 0.56 (Alabama sites). A comparison was made to determine whether the sampling frequency was representative of the trends in a complete dataset: FG and OM concentrations of the 2016 dataset, which included all days, were contrasted with





those of the 1:3 days dataset (Sect. S7). While the lack of daily samples added noise to the data, there was no apparent bias in OM or FG concentrations. Further clues into the changes in atmospheric chemistry leading to the overall decrease in OM can be drawn from the concentrations of FGs measured using FT-IR spectrometry.

### 3.2 Multi-year trends of OM composition using functional groups: Observations

5    The FG composition in the SE U.S. was dominated by aCH and COOH, each contributing between 24 and 35% of OM. aCOH contributed between 16 and 30%, while oxOCO and naCO contributed less than 15% over all sites and years (Figure 3). The FGs did not decline uniformly with OM over the 2011–2016 time period; rather, the composition changed, suggesting a change in dominant sources and atmospheric processes. As shown in Figure 3, the contributions of the more oxygenated FGs, COOH and oxOCO, decreased, while those of the moderately oxygenated FGs, aCOH and naCO, became

10   slightly enriched (aCH did not change significantly).

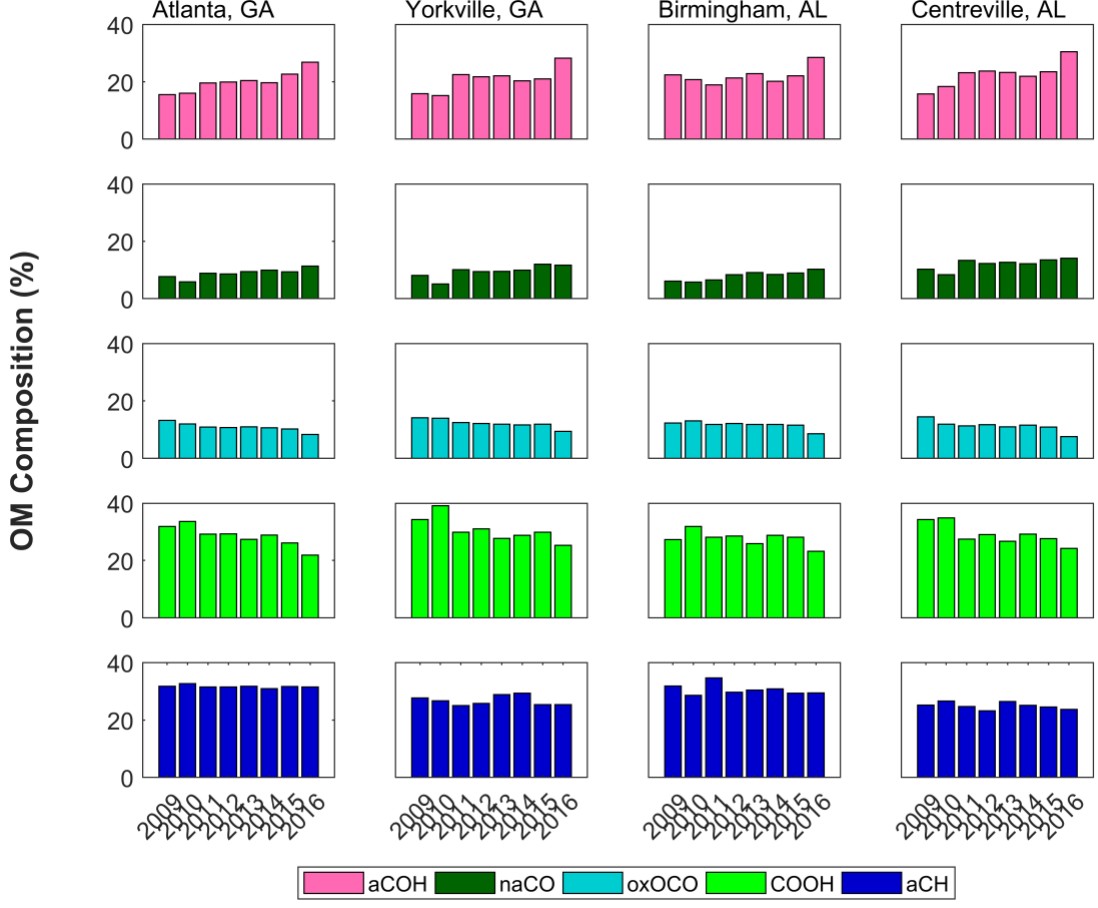

**Figure 3. Annual median FG fractions of OM composition as quantified via FT-IR spectrometry. Column-wise subplots correspond to individual sites.**



Within the annual median trends in FG/OM, the 2016 annual median OM composition is notably different from that in 2015. 2016 samples were freshest when analyzed via FT-IR spectrometry; to evaluate possible analytical causes of this difference, storage or sample handling losses were considered (Boris et al., 2019; Sect. S1.2). Given that the observed trends in FG abundances continued into the 2018 extended dataset (January and February; Sect. S5) and results of re-analyzing spectra after 1–2 years of storage did not explain this behavior, these trends appear to be robust. 2009–2010 FG contributions likely differed from subsequent years due to the 2008 recession (Sect. 3.1).

The trends in FG abundance were also mirrored in the FT-IR spectra: changes of the absorbance regions associated with each quantified FG in annual median spectra at each site demonstrated similar patterns (Figure 4; Sect. S5).

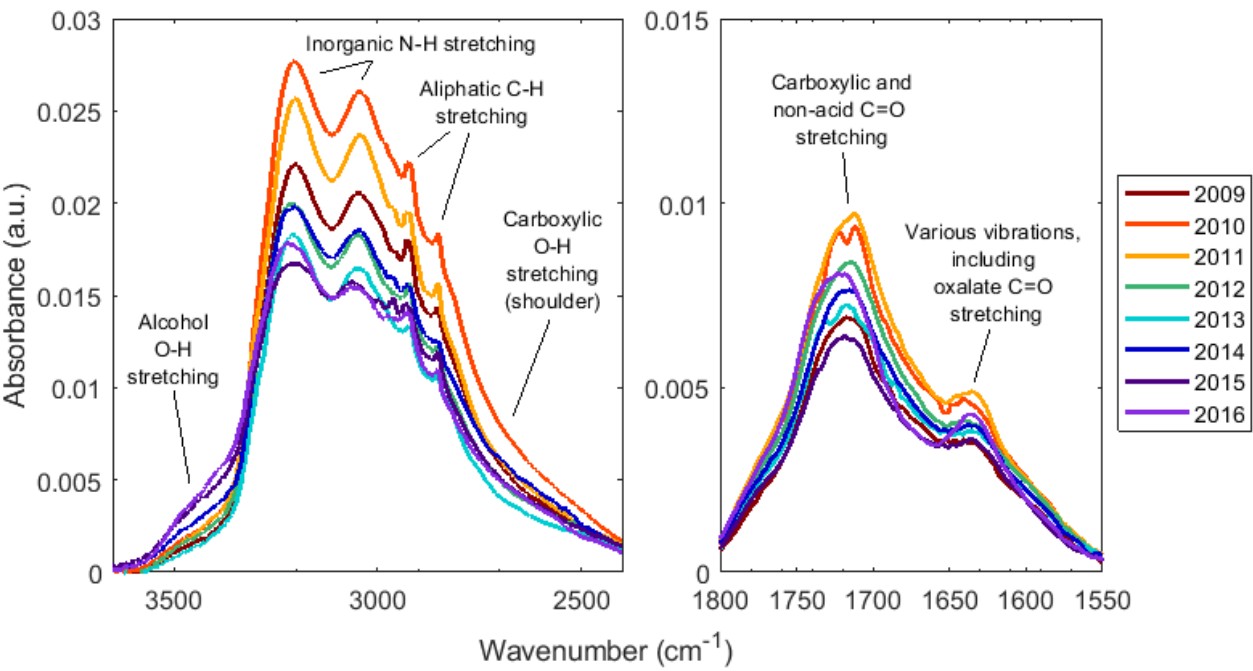

**Figure 4. Annual median baseline corrected spectra of samples collected at JST. Colored lines, as indicated in the legend, demonstrate a progression of absorbance over time. For baseline correction and blank spectrum subtraction details, see Sect. S5).**

An absorbance peak at approximately 3500 cm$^{-1}$, corresponding to the aCOH O-H stretching absorption, increased, while the shoulder between ~2650 and 2900 cm$^{-1}$ corresponding to COOH O-H stretching decreased in absorbance. In the right subplot of Figure 4, the C=O stretching region (from naCO, but also COOH groups) absorbance increased from 2009 to 2011 and demonstrated a general decline after 2011. Similarly, the absorbance due to inorganic N-H stretches and aCH (the sharp peaks between ~2800 and 3000 cm$^{-1}$ correspond to aCH C-H stretching absorption) were greatest in 2010, and that due to vibrations including oxOCO C=O stretching at ~1630 cm$^{-1}$ were greatest in 2011. Note that although their absorption spectra are quite different (Hay and Myneni, 2007), COOH and oxOCO are related: the origin of oxOCO in OM is likely



deprotonation of COOH and coordination with another cation such as $Na^+$ or $NH_4^+$ (Ortiz–Montalvo et al., 2014) or within metal complexes (Sorooshian et al., 2013). These same trends were also generally shown by spectra from other sites (Sect. S5).

Multi-annual, regional decreases in the relative contributions of COOH and oxOCO were offset by increases in the naCO

5 and aCOH contributions; in absolute terms, the concentrations of naCO and aCOH did not trend significantly (Table 2). Trends in FG measurements by season and site helped to elucidate influences on atmospheric chemistry. Site-specific trends demonstrated in general that urban area samples changed most over time (with the exception of oxOCO), likely reflecting the availability of carbon sources and/or oxidants near cities, although trends were often similar in magnitude between sites.

**Table 2. Trends in concentrations (µg m⁻³ yr⁻¹) and FG/OM (%OM yr⁻¹) measured using FT-IR spectrometry. Trends**
10 **were calculated using Theil–Sen regression and expressed 2011–2016 (Sect. 2.8). Confidence limits are bootstrapped at 95 % confidence. Slopes in bold are greater than the confidence limits.**

| Variable | Sites | Seasons | Conc. Slope ± Confidence Limit (µg m⁻³ yr⁻¹) | FG/OM Slope ± Confidence Limit (%OM yr⁻¹) |
|---|---|---|---|---|
| aCH | All | All | **-0.03** ± 0.02 | -0.2 ± 0.3 |
| aCH | CTR | All | -0.01 ± 0.03 | -0.1 ± 0.4 |
| aCH | YRK | All | -0.02 ± 0.04 | 0.0 ± 0.4 |
| aCH | JST | All | -0.03 ± 0.05 | -0.2 ± 0.5 |
| aCH | BHM | All | **-0.07** ± 0.06 | -0.4 ± 0.6 |
| aCH | All | Winter | -0.02 ± 0.04 | -0.1 ± 0.7 |
| aCH | All | Spring | -0.01 ± 0.04 | -0.1 ± 0.5 |
| aCH | All | Summer | **-0.08** ± 0.06 | **-0.5** ± 0.4 |
| aCH | All | Autumn | -0.02 ± 0.05 | -0.1 ± 0.6 |
| COOH | All | All | **-0.04** ± 0.03 | **-0.9** ± 0.4 |
| COOH | CTR | All | -0.02 ± 0.04 | **-0.8** ± 0.5 |
| COOH | YRK | All | -0.04 ± 0.05 | **-0.9** ± 0.8 |
| COOH | JST | All | -0.04 ± 0.05 | **-1.1** ± 0.7 |
| COOH | BHM | All | -0.06 ± 0.06 | -0.7 ± 0.9 |
| COOH | All | Winter | -0.01 ± 0.03 | 0.13 ± 0.8 |
| COOH | All | Spring | -0.04 ± 0.04 | **-1.4** ± 0.7 |
| COOH | All | Summer | **-0.12** ± 0.07 | **-1.4** ± 0.6 |
| COOH | All | Autumn | -0.02 ± 0.04 | -0.5 ± 0.7 |
| oxOCO | All | All | **-0.024** ± 0.007 | **-0.59** ± 0.13 |
| oxOCO | CTR | All | **-0.019** ± 0.012 | **-0.71** ± 0.25 |
| oxOCO | YRK | All | **-0.021** ± 0.014 | **-0.52** ± 0.24 |
| oxOCO | JST | All | **-0.022** ± 0.015 | **-0.54** ± 0.23 |



| Variable | Sites | Seasons | Conc. Slope $\pm$ Confidence Limit ($\mu g\ m^{-3}\ yr^{-1}$) | FG/OM Slope $\pm$ Confidence Limit (%OM $yr^{-1}$) |
|---|---|---|---|---|
| oxOCO | BHM | All | **-0.037** $\pm$ 0.018 | **-0.5** $\pm$ 0.3 |
| oxOCO | All | Winter | **-0.014** $\pm$ 0.012 | -0.3 $\pm$ 0.3 |
| oxOCO | All | Spring | **-0.024** $\pm$ 0.014 | **-0.82** $\pm$ 0.28 |
| oxOCO | All | Summer | **-0.041** $\pm$ 0.017 | **-0.47** $\pm$ 0.23 |
| oxOCO | All | Autumn | **-0.021** $\pm$ 0.014 | **-0.61** $\pm$ 0.20 |
| naCO | All | All | -0.003 $\pm$ 0.008 | **0.45** $\pm$ 0.21 |
| naCO | CTR | All | 0.004 $\pm$ 0.021 | 0.3 $\pm$ 0.4 |
| naCO | YRK | All | 0.000 $\pm$ 0.016 | 0.4 $\pm$ 0.4 |
| naCO | JST | All | 0.008 $\pm$ 0.017 | **0.5** $\pm$ 0.3 |
| naCO | BHM | All | -0.001 $\pm$ 0.019 | 0.5 $\pm$ 0.5 |
| naCO | All | Winter | -0.014 $\pm$ 0.016 | -0.3 $\pm$ 0.5 |
| naCO | All | Spring | 0.018 $\pm$ 0.019 | **0.8** $\pm$ 0.4 |
| naCO | All | Summer | 0.000 $\pm$ 0.021 | **0.7** $\pm$ 0.4 |
| naCO | All | Autumn | 0.009 $\pm$ 0.020 | 0.4 $\pm$ 0.4 |
| aCOH | All | All | -0.009 $\pm$ 0.019 | **1.16** $\pm$ 0.29 |
| aCOH | CTR | All | 0.02 $\pm$ 0.04 | **1.2** $\pm$ 0.5 |
| aCOH | YRK | All | 0.00 $\pm$ 0.04 | **1.0** $\pm$ 0.6 |
| aCOH | JST | All | 0.02 $\pm$ 0.04 | **1.3** $\pm$ 0.5 |
| aCOH | BHM | All | -0.01 $\pm$ 0.04 | **1.0** $\pm$ 0.7 |
| aCOH | All | Winter | 0.00 $\pm$ 0.03 | 0.8 $\pm$ 0.8 |
| aCOH | All | Spring | 0.03 $\pm$ 0.04 | **1.5** $\pm$ 0.7 |
| aCOH | All | Summer | 0.00 $\pm$ 0.05 | **1.7** $\pm$ 0.6 |
| aCOH | All | Autumn | 0.01 $\pm$ 0.04 | **0.7** $\pm$ 0.5 |

Larger compositional (FG/OM) changes occurred in spring and/or summer, dominated by the larger magnitude declines in COOH and oxOCO (Table 2). In contrast, the positive changes in naCO and aCOH concentrations were steepest in the springs, followed by autumns, although not significant (Table 2). Monthly medians supported the seasonalities of naCO and aCOH trends (Sect. 3.5). This demonstrated that the loss in more oxygenated OM concentrations was mainly in summer and spring, while any trend in the less oxygenated OM concentrations occurred in the shoulder seasons.

### 3.3 Multi-year trends of OM composition using functional groups: Interpretation

These annual and seasonal FG measurements, mirrored by the FT-IR spectra themselves, concisely demonstrate trends in OM composition. They provide observations useful in identifying its sources and formation processes, and tie together studies of SE U.S. aerosol with more limited scopes in time or chemical analysis. The following paragraphs provide


interpretation of the trends in FG contributions to OM informed by the current results, as well as a variety of laboratory, modelling, and campaign-based studies relevant to the SE U.S. OM composition. While FGs are not specific to any particular emission sources or processes, their variations can provide insight into atmospheric processes. Therefore, past FT-IR spectrometry source attribution studies with factors from positive matrix factorization (PMF) are considered in particular

to help guide the discussion of trends.

### 3.3.1 Interpretation of multi-year decrease in COOH and oxOCO contributions to OM

The most distinct trend in FG composition is a decline of COOH and oxOCO abundance over time (-0.9% COOH/OM yr[-1]; -0.59 % oxOCO/OM yr[-1]; Table 2). Since oxOCO has been studied less in previous work, we focus here on COOH. COOH is certainly associated with highly oxidized material; aerosol mass spectrometry work in SE U.S. has demonstrated a strong

relationship between low molecular mass species with COOH and more oxidized oxygenated organic aerosol (Chen et al., Submitted). Past FT-IR spectrometry studies have shown that COOH contribution to OM indicates more aged combustion emissions relative to other factors, and especially fossil fuel combustion (at Whistler, British Columbia: ~30 %, Schwartz et al., 2010 and ~45 %, Takahama et al., 2011; ~42 % in South Central California, Liu et al., 2011; ~33 % in Houston area oil combustion, Russell et al., 2009; and ~40–45 % in a heavily forested region of Finland, Corrigan et al., 2013). This suggests

the likely contribution of anthropogenic emissions to the declines in OM (and COOH and/or oxOCO). The two FT-IR factors that were found in SE U.S. aerosol to be elevated in COOH contributions, aged combustion and mixed organic aerosol, were also correlated with $SO_4^{2-}$ concentrations (Liu et al., 2018). This further suggested a shared oxidation process and/or fossil fuel combustion source between the declining OM and $SO_4^{2-}$. In the present work, FT-IR COOH and oxOCO concentrations were correlated with those of $SO_4^{2-}$ ($R^2$=0.48 and $R^2$=0.54, respectively), and both species declined greatest at

an urban site (BHM). Based on this and previous FG measurements, the decreasing trends in COOH and oxOCO can be attributed to regionally declining emissions of anthropogenic VOCs and/or $SO_2$.

Other measurements and model results, summarized below, elucidate three probable mechanisms that link the parallel declines in oxidized OM concentrations with those in anthropogenic VOC and/or $SO_2$ emissions. First, photo-oxidation of anthropogenic VOCs, which accounts for ~18–30 % of OM in the SE U.S., can directly explain some portion of the decline in oxidized OM (Blanchard et al., 2013a; Kim et al., 2015; Mao et al., 2018). This anthropogenic portion is typically not

discussed, with the exception of the works of Ridley et al. (2018) and Blanchard et al., (2016). Discussions have instead focused more on the contribution of oxidized isoprene (biogenic) emissions, which accounts for ~18–36 % of OM (summer 2013; Budisulistiorini et al., 2015; Xu et al., 2015b), and introduces a second probable mechanism for OM decline. While anthropogenic emissions have themselves been declining, isoprene OM is suggested to be declining, not due to direct decrease in isoprene emissions, but indirectly due to mechanisms related to decreasing $SO_2$ emissions (Goldstein et al., 2009;

Malm et al., 2017; Marais et al., 2017; Nguyen et al., 2014; Xu et al., 2015b). The precise mechanism for this is uncertain, but relates to $SO_4^{2-}$ concentrations (an oxidation product of $SO_2$): while some research supports a pathway related to particle acidity and/or aerosol water content (Chen et al., Submitted; Marais et al., 2017; Sareen et al., 2016; also see Sect. S1.1





discussing the possible role of aqueous oxidation), other studies suggest that isoprene OM declines are less associated with these factors (Budisulistiorini et al., 2015), and more directly associated with $SO_4^{2-}$ concentrations and the formation of organosulfates (Schindelka et al., 2013; Xu et al., 2015b). One recent study suggests, however, that the impact of $SO_4^{2-}$ on these observed OM concentration declines could be over-estimated (Zheng et al., 2020). Third, the oxidation pathways of

anthropogenic or monoterpene VOCs may be affected by $SO_4^{2-}$ concentration declines, with parallel pathways to the suggested decline in isoprene-derived OM. Overall, we suggest that the contribution of secondary OM, composed partially of COOH and oxOCO, is decreasing directly due to anthropogenic VOC emissions declines and/or indirectly due to $SO_2$ emissions declines.

### 3.3.2 Interpretation of multi-year increase in aCOH and naCO contributions OM

Another trend in FG composition is the relative enhancement in the aCOH and naCO over time (+1.2 % aCOH/OM $yr^{-1}$; +0.45 % naCO/OM $yr^{-1}$; Table 2). While the trends in terms of absolute concentrations of aCOH and naCO are not significant (Table 2), they are clearly distinct from those in the more oxygenated FGs. This trend represents the emergence of a new SE U.S. OM composition, and likely a shift in the dominant sources and/or processes contributing to OM. It opposes the decline in abundance observed for most measured anthropogenic pollutants, including $NO_x$, $SO_2$, and the

contribution of $SO_4^{2-}$ to $PM_{2.5}$. An aggregation of relevant field and lab observations from the literature regarding the trends in aCOH and naCO point to oxidized monoterpenes as a source of carbon and possible chemical formation processes by low $NO_x$ oxidants, as described below.

A relationship of aCOH and naCO with oxidized biogenic emissions was reported in prior FT-IR spectrometry work from the SE U.S. (Liu et al., 2018). Liu et al. demonstrated agreement with the strong aCOH and naCO contributions to OM found

in other FT-IR spectrometry biogenic factors from mainly evergreen-dominated forests (Corrigan et al., 2013; Schwartz et al., 2010; Takahama et al., 2011). The seasonality of aCOH and naCO concentrations may also support a monoterpene source: slopes were steepest during the spring followed by autumn (Table 2), coincident with events driving monoterpene emissions, including bud break and needle growth in the spring and leaf fall in the autumn (Geron and Arnts, 2010; Kim, 2001). (Note that we focus here on the seasonal behavior of the less oxygenated FGs independently of the overall trend in

OM by looking at changes in concentrations rather than in FG/OM.) Particular mechanisms for preferential formation of aCOH and naCO have not been suggested, but monoterpene oxidation does result in formation of these FGs (Yu et al., 1998). Less-oxidized oxygenated organic aerosol (LO-OOA; a factor identified using high resolution aerosol mass spectrometry) has also been strongly associated with monoterpenes in the SE U.S. (Xu et al., 2018; sesquiterpenes may also be related); this and other literature has identified the majority of recent SE U.S. OM as secondary products from

monoterpene oxidation (Xu et al., 2015b; Zhang et al., 2018).

The production of secondary biogenic OM also varies with $NO_x$ concentrations in the SE U.S. (Liu et al., 2018; Xu et al., 2018; Zhang et al., 2018), perhaps explaining some of the relative enhancement in less oxygenated FG abundance ($NO_x$ concentrations could affect oxidant and oxidation product distributions; Ziemann and Atkinson, 2012). Recent SEARCH



observations have supported the continued or even enhanced availability of oxidants that are influential at low $NO_x$ concentrations, including $O_3$ and thus $^\bullet OH/HO_2^\bullet$. While the concentrations of $NO_x$ in the SEARCH network have declined (the median decreased from 0.3 ppb in 2011 to 0.2 ppb in 2016), $O_3$ concentrations increased (from a median of 31 ppb in 2011 to 34 ppb in 2016). Furthermore, the summary of SE U.S. modelling studies provided by Mao et al. (2018) suggests

that the SE U.S. is shifting from $NO_x$-dominated to $O_3$-dominated (and thus $^\bullet OH$) chemistry. Low-$NO_x$ reactions, including production of additional HOM-$RO_2$ via autoxidation or HOM accretion, could lead to condensation of volatile aCOH and naCO-containing molecules (Pullinen et al., 2020; Pye et al., 2019; Ziemann and Atkinson, 2012). A recent lab-in-the-field study demonstrated that $O_3$ concentration was the only variable among many with which secondary OM mass from α-pinene oxidation was strongly correlated in ambient SE U.S. air, supporting this hypothesis (Xu et al., 2018). However, the

relationship between $NO_x$ and secondary OM production is not straight-forward: $NO_x$ is a key reactant in $O_3$ and $^\bullet OH$ formation, and also organonitrates during the daytime, all of which contribute to OM concentrations in the SE U.S. (Pye et al., 2015; Xu et al., 2015a). A recent body of literature has worked to predict the direction of the effect of declining $NO_x$ emissions on OM formation (summarized by Liu et al., 2018); some of this research supports a causal relationship between $NO_x$ concentrations and the increasing trend in aCOH and naCO fractions observed in the present work.

$O_3$ increases could alternatively be responsible for enhanced OM fractions of aCOH and naCO independent of $NO_x$ concentrations. Matching the seasonalities of the trends in aCOH and naCO concentrations (Table 2; FG/OM seasonalities are likely dominated by COOH and oxOCO), the concentrations of $O_3$ increased during springs and autumns 2011–2016 (while decreasing in summers). The rise in autumn $O_3$ concentrations has been attributed to drought conditions and meteorological patterns in the SE U.S. that are not directly linked to long-term anthropogenic $NO_x$ declines (Zhang and

Wang, 2016; -0.1 ppb yr$^{-1}$ in SEARCH data). As stated above, aCOH and naCO can be oxidation products of monoterpenes (Liu et al., 2018; Sax et al., 2005; Yu et al., 1999), emissions which are relatively abundant during the spring through autumn (Geron and Arnts, 2010). Even while reaction of monoterpenes with $O_3$ is slower than $^\bullet OH$ and $NO_3^\bullet$ at peak oxidant concentrations (e.g., Ziemann and Atkinson, 2012), oxidation by $O_3$ can be sustained when these other oxidants are found in lower abundance (Peräkylä et al., 2014) and generate secondary OM with lower volatility (Watne et al., 2017).

There are additional alternative hypotheses that may contribute to the observed trends in the contributions of aCOH and naCO to OM. As outlined by several studies of the SE U.S. (Ayres et al., 2015; Mao et al., 2018; Xu et al., 2015a; Zhang et al., 2018), organonitrates could account for a substantial fraction of OM in the region, even at low $NO_x$ concentrations, and are known to hydrolyse efficiently to give aCOH-containing products. However, since $NO_3^\bullet$ concentrations would have declined alongside $NO_x$ concentrations, it is unlikely that organonitrate hydrolysis can explain the increase in the prominence

of aCOH and naCO ($NO_3^\bullet$ at night and NO during the day are likely responsible for SE U.S. organonitrate formation; Xu et al., 2015a). Smoke is another plausible contributor of the measured aCOH and naCO since emissions from prescribed burns, wildfires, and residential woodburning contribute substantially to SE U.S. aerosol (Peräkylä et al., 2014), and aCOH and naCO have been associated with biomass burning (Corrigan et al., 2013; Liu et al., 2018, 2009; Russell et al., 2011; Takahama et al., 2011). However, neither the enhancement in smoke over the study period (prescribed and wildland fires





may actually have decreased in acreage; Coalition of Prescribed Fire Councils, 2012, 2015, 2018) nor a clear enhancement in these FGs during fire periods could be demonstrated (Sect. 3.6; Sect. S1.3), possibly related to strong variability with other influences on the FG contributions to biomass burning FT-IR factors (Russell et al., 2011). Likewise, residential woodburning was concluded not to be a responsible factor because the expected wintertime increase was not in line with FG

trend seasonality. Isoprene epoxydiol (IEPOX) reactions could also generate aCOH groups in particular (Chan et al., 2010; D'Ambro et al., 2019); however, the increase in aCOH concentration is not strong in summer, which is when IEPOX chemistry is most pronounced (Budisulistiorini et al., 2016), and prior modeling work (Marais et al., 2017) and FT-IR spectrometry (e.g., Corrigan et al., 2013) work suggest that that isoprene emissions are not likely to be responsible for this growing fraction of OM.

**3.4 Annual and seasonal trends in the OM/OC Ratio**

Seasonal and annual median values of OM/OC varied between ~2.0 and 2.4 (Figure 5a), greater than some typical OM/OC values used for TOR OC conversion in many networks (e.g., 1.8 or 1.4; (Blanchard et al., 2013b; Malm et al., 2017), but similar to those used in more current work (Marais et al., 2017). The FT-IR-derived OM/OC values are similar to those measured in previous work on SE U.S. aerosol: an OM/OC of ~2.15 was observed by El–Zanan et al. (2009) using masses of

solvent extracts and a mass balance method; ~1.9 was observed using re-suspension and high resolution aerosol mass spectrometry by Sun et al. (2011b); ~1.8–2.2 was observed by Hand et al. (2019) for 2009–2016 IMPROVE network data using a multiple regression method; and 2.24 was observed using direct aerosol mass spectrometry by Kim et al. (2015). Especially with respect to OM/OC values, it must be noted that some molecular bonds are not captured using FT-IR spectrometry: for example, the organic $SO_4^{2-}$ and $NO_3^-$ groups that might be particularly relevant in discussions of SE U.S.

aerosol are not included in the OM contribution. However, the majority of OM and OC masses are captured (Boris et al., 2019).





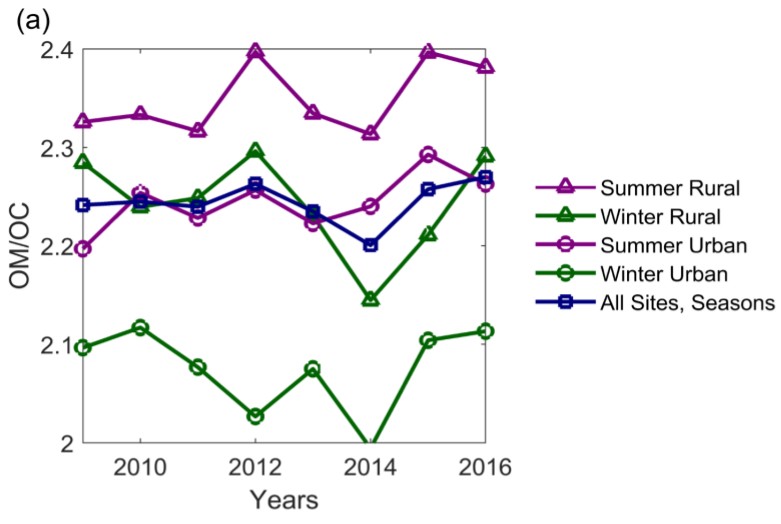

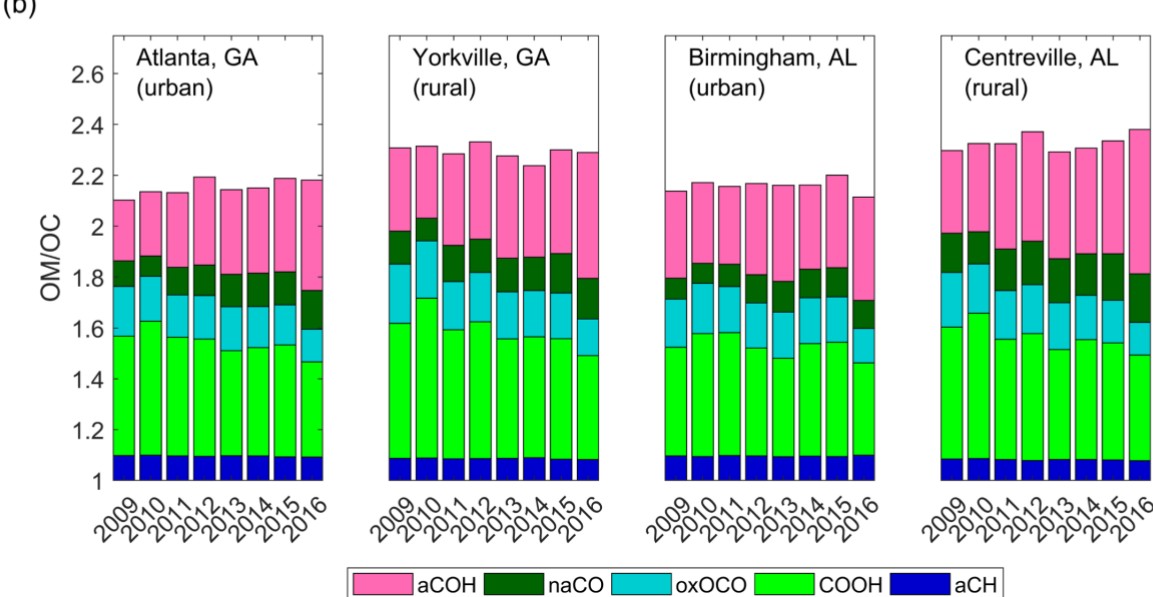

**Figure 5. OM/OC, expressed as (a) seasonal median ratios at urban and rural grouped sites and (b) annual median functional group contributions to OM/OC ratios at each site.**

During the summertime, when photochemistry is enhanced, the degree of OM oxidation is generally expected to be greater than in months with less solar insolation (Blanchard et al., 2008). Likewise, a greater degree of oxidation is anticipated at rural sites due to aging of urban pollutants. Median OM/OC ratios were greater in summers than in winters, and also greater at rural than at urban sites, in agreement with these expectations (Figure 5).

The competing impacts of the changes in FG concentrations resulted in a trend in the OM/OC that was not statistically significant ($+0.002 \pm 0.008$ yr$^{-1}$ 2011–2016; Figure 5b; Sect. S2). In parallel to absolute trends of FGs (Sect. 3.2), material





contributed to OM/OC by COOH and oxOCO declined during this period, while there was an enhancement in aCOH and naCO concentrations (the aCH contribution was statistically unchanged; $+0.002\pm0.003$ yr$^{-1}$). Trends in OM/OC for each of the sites and seasons were also not significant. Using a similar measurement technique, Bürki et al. (2020) showed no trend in OM/OC for annual medians of 2011 and 2013 IMPROVE network sites, including at St. Marks, Florida. However, the FG

contributions did not appear to vary, in contrast to the findings of the current calibration models and SEARCH dataset.

Differing from FG OM/OC trends, the work of Hand et al. (Hand et al., 2019) using a multiple linear regression method demonstrated an increase in OM/OC of SE U.S. aerosol between 2011 and 2014, and a decrease between 2014 and 2016. These trends were attributed either to changes in the character of OM or OC that were ubiquitous throughout the U.S. and over all seasons, to analytical challenges in the various instrumental/collection methods, or calculation/algorithm challenges

of reconstructed fine mass composition. The trend in residual OM / TOR OC also increased with time ($+0.053\pm0.015$ yr$^{-1}$ between 2011 and 2016; see Sect. 2.6 for residual OM definition), further suggesting that FG OM/OC provide different results from OM/OC determined using TOR OC and IMPROVE data. Using the residual OM and TOR OC concentrations, the median value of OM/OC was also only 1.65, lower than anticipated based on other methods (see above; e.g., El–Zanan et al., 2009; Hand et al., 2019).

Efforts to apply multiple linear regression methods to 2011–2016 SEARCH data similar to Hand et al. (2019) suggested a systematic discrepancy in the results: derived regression coefficients were not physically meaningful with respect to expected chemistry (Simon et al., 2011). Possible explanations for the lack of physically meaningful results included volatilization of nitrate (although nitrate was expressed as the sum of the Teflon and backup nylon filters; Edgerton et al., 2005b), inadequate accounting for particle water (although this was approximated as in Part 1 of this work, Boris et al.,

2019) and/or TOR OC concentration abnormalities due to hardware degradation or changes in methodology (Hand et al., 2019). These limitations notably did not prohibit calculation of OM/OC using FT-IR spectrometry measurements, since the OM and OC are estimated directly from FT-IR spectra of samples and collected chemical standards.

In addition to OM/OC, FG measurements were used to calculate the carbon oxidation state (OS$_C$) as a metric of degree of oxidation, demonstrating a lack of trend, as with OM/OC ratios. Values of OSc were within the range of semi-volatile or

low-volatility oxygenated OM but were more oxidized than aerosol from urban sources or biomass burning (Kroll et al., 2011; Sect. S3).

**3.5 Trends and changes in seasonal OM composition**

Summertime aerosol carbon in the SE U.S. is predominantly from biogenic sources, with particularly strong seasonality in isoprene emissions (Guenther et al., 1994; Xu et al., 2015a). The greater photochemistry and stagnant conditions in the

summertime in the SE U.S. are also favorable for enhanced secondary OM production, with high OM concentrations originating from urban as well as remote sources (Hidy et al., 2014b; supporting meteorological observations are summarized in Sect. S13; select wind roses are shown in Sect. S6). Absolute FG OM concentrations were greater during the



long SE U.S. warm season than in colder months (Figure 6), despite the higher boundary layer during the warmer months in the SE U.S. (Seidel et al., 2012). While the seasonality of all FG concentrations generally matched that of OM, with greater abundances in the summer/autumn than winter for all years, the relative contribution of COOH to OM was significantly greater in the summer (Figure 6; Fig. S9). The greater summertime OM concentration and COOH fraction of OM support

enhanced oxidation and isoprene emission as a prominent source of COOH in warmer months (in line with Sect. 3.3.1 findings).

In contrast, markers of fossil fuel combustion and wood burning have been measured at greater concentrations in the winter and spring in the SE U.S. (Blanchard et al., 2014; Sun et al., 2011a). FT-IR spectrometry results mirrored these observations: the relative contribution of aCH to OM was greatest in the wintertime (Fig. S9), and the urban/rural ratios of OM

concentrations were greater in the cold months (~1.2–1.4 December–February) than the warm months (1.0–1.2 June–August; Georgia sites; Fig. S10). Indicative of a seasonality of the carbon emissions instead of secondary production (Sun et al., 2011a), the same seasonality was not observed in urban/rural ratios of $SO_4^{2-}$ concentration (Fig. S12). The impact of aged of combustion emissions was also demonstrated by the elevated cold season particle $NO_3^-$ concentrations observed, (2013–2016 monthly medians; e.g., Fig. S16; Kleeman et al., 2005). These results suggest that anthropogenic/combustion sources were

more prominent during the colder months in the SE U.S., and that the contribution of the FG aCH was indicative of those sources.





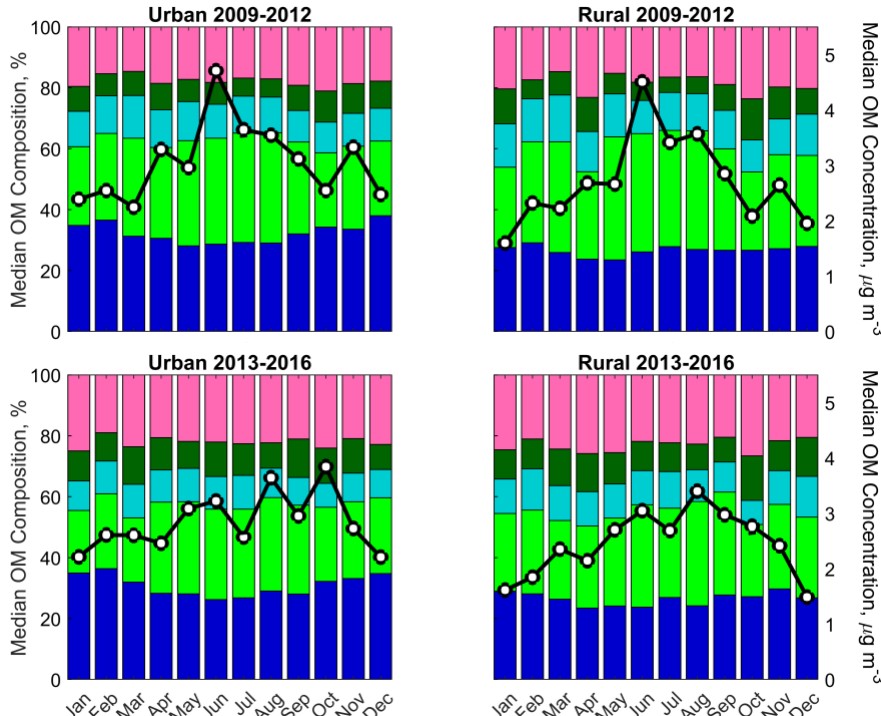

**Figure 6. Monthly FG composition and OM concentration during 2009–2012 (top) and 2013–2016 (bottom), urban (left) and rural (right) subcategories. Stacked color bars represent the monthly median FG contributions to OM concentration (see Figure 2 for FG legend), while black lines with white markers represent the monthly median OM concentrations.**

An annual peak in monthly median OM concentrations was observed in June in the earlier years, but late summer/early autumn months in the later years (Figure 6; with notable variation between years: 2009 and 2011 have the most prominent June peaks, but 2016 also has a small June peak, and the most prominent late summer/autumn peak was in 2013; Fig. S8). The June peak was also observed in all FG concentrations 2009–2012, while greater aCOH, naCO, and aCH concentrations were observed in autumn 2013–2016 (Fig. S9). Since the seasonality of monoterpene-related OM is much less pronounced than isoprene OM in the SE U.S. (as measured via aerosol MS; Xu et al., 2018), this seasonality of aCOH and naCO, in particular, supports the hypothesis put forward in Sect. 3.3.2 (which was based on other FT-IR spectrometry work; Russell et al., 2011) that these FGs are predominantly attributable to monoterpene oxidation products.

**3.6 2016 daily dataset: observations of warm season stagnation and cold-season combustion emissions**

FT-IR spectra were collected for all sites and each day in 2016 until site closure (Sect. 2.1). This rich dataset allowed identification of trends across seasons and additional comparison between fire/non-fire periods. Weekly trends were also observed in OM and FG concentrations using this higher resolution data. While weekend concentrations were lowest only





for $NO_x$, species including all FGs, OM, TOR OC and EC, x-ray fluorescence K, and CO were lowest in concentration on a midweek day, and generally Tuesdays (medians during winters and summers, 2013–2016, at urban and rural sites). This observation may be related to feedbacks with regional rainfall (Bell et al., 2008), although there was no clear pattern between weekly concentration minima and hygroscopicity of species (Sect. S14).

Lower than normal precipitation and higher than normal temperatures in 2016 led to stronger drought conditions and wildfires, especially during the latter part of the year, although prior years were also dry in the SE U.S. (Konrad II and Knox, 2018). OM concentrations were dynamic, with build-up and flushing out of material on a more rapid timescale in the cold season than in the warmer months. The source of the dynamic OM concentrations appeared to differ by season as well, with smoke and secondary OM the most likely controls on OM variation in the cold and warm seasons, respectively.

The fractional contributions of FGs to total OM concentrations differed between samples for which smoke was visible in satellite images versus those with no indication of smoke impact (evaluated at the 95 % confidence level; Sect. S1.3). In samples with suspected fire impact, aCH/OM was significantly higher, naCO/OM and aCOH/OM were not significantly different, and COOH/OM and oxOCO/OM were significantly lower than in samples with no indication of fire impact (Fig. S1). This FG "fingerprint" for smoke appears to be unique to the SE U.S., with somewhat different FG ratios observed in

smoke-impacted samples elsewhere (Corrigan et al., 2013; Russell et al., 2011). Greater x-ray fluorescence K concentrations were also observed in samples with suspected fire impact, supporting the conclusion that observed differences between "fire" and "non-fire" samples were not due to other factors such as seasonality (Fig. S1). While it is possible that the observed lack of difference in naCO and aCOH contributions to OM in 2016 "fire" versus "non-fire" samples is attributable to the overlapping contribution from other OM sources and/or processes of these two FGs, it may alternatively suggest that

these species are not contributed in particular by smoke from wildfires and/or prescribed burns. The latter would contradict the alternative hypothesis that the multi-annual increases in naCO and aCOH concentrations are due to smoke, despite the relationships identified in prior FT-IR factor analysis work (Sect. 3.3.2).



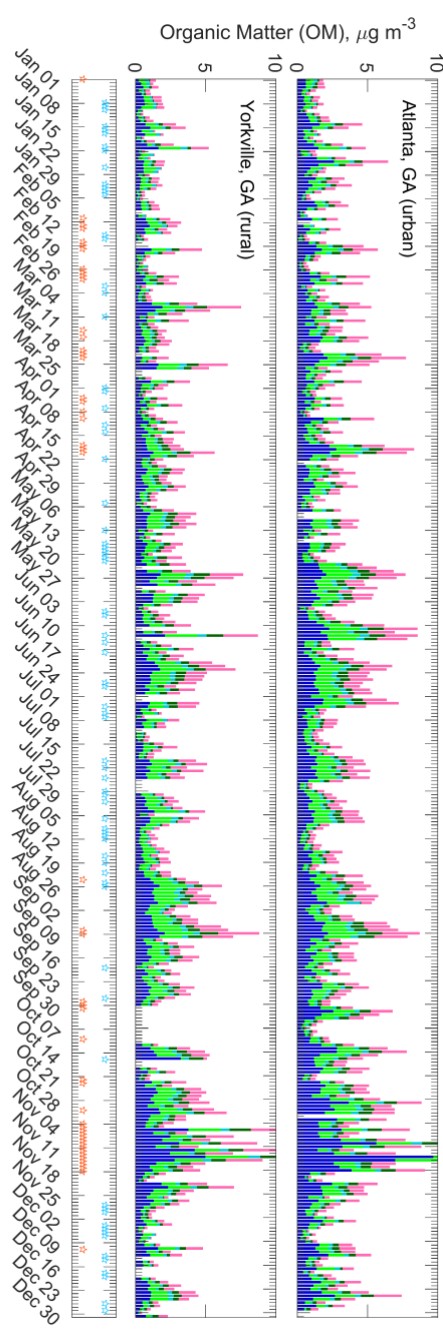

**Figure 7. Absolute OM and FG concentrations during 2016 for the Georgia urban (top) and rural (center) sites. Some values (three at each site) extend beyond the vertical axis scale so that variation can be observed within low and typical OM concentration ranges. Events related to OM concentration dynamics are represented in the bottom plot:**



**precipitation events correspond to blue markers; fires correspond to orange markers. See Figure 2 for legend FG colors.**

Winter/spring (1 January through 20 April; Figure 7) was distinct in that smoke plumes visible in satellite imagery were often coincident with high OM concentrations. Prescribed burning likely explains this observation: it is widespread in the SE U.S. between January and April on days when conditions are appropriate (Larkin et al., 2014). Events in precipitation and related meteorological variables typically coincided with decreases in OM concentrations after elevated OM periods.

In contrast, during the late spring/summer (21 April – 25 August), the contribution of OM from prescribed burning sources is expected to be minimal because burns of most types are not permitted in Alabama and Georgia (Alabama Forestry Commission, 2019; Georgia Environmental Protection Division, 2020). Convective activity in the summer also brings regular rain (Diem and Mote, 2005), and local wildfires are uncommon relative to other seasons (Larkin et al., 2014). Visible satellite imagery showed no smoke plumes during the 2016 late spring/summer. Summer OM events were likely caused by secondary OM formation with carbon originating from biogenic sources (Sun et al., 2011a), as discussed in Sect. 3.5, or from aged smoke transported from wildfires in the Western U.S. The high OM events were longer in duration than those in the winter/spring: springtime events were on the order of a few days, and were punctuated by frontal passages, while summertime events sometimes lasted 1.5 weeks, interrupted by less frequent meteorological events (the latter was also observed by (Liu et al., 2018).

Periods of OM build-up also occurred during the autumn (26 August – 31 October), but smoke plumes were again visible using satellite imagery, indicating that local fires may have contributed to OM.

Wood smoke appeared to be ubiquitous during the winter season in the SE U.S. (defined here as 31 October to 31 December), based on observed elevated aCH/OM in the winters (Fig. S18 and supported in prior work; Chen et al., 2012), although the overlap of other combustion sources such as vehicle emissions is probable. Severe wildfires in the southeastern Appalachian Mountains occurred at the end of October and into mid-November, due in part to a drought (Konrad II and Knox, 2018), resulting in the highest seasonal OM concentrations observed in 2016. In agreement with the lack of smoke "fingerprint" in the absolute concentrations of FGs, in the 2009–2016 1:3 day dataset, no correlations between FG concentrations and non-soil K were observed ($0.10 < R^2 < 0.17$; non-soil K calculated from XRF K and Si concentrations; Blanchard et al., 2016a). Additional precision in identifying smoke impact on FG composition may be possible by performing factor analysis (Russell et al., 2011).

## 4 Conclusions

OM concentrations declined at all SEARCH sites from 2011 to 2016, due to decreasing concentrations of COOH, oxOCO, and aCH, while the relative contributions to OM of aCOH and naCO increased. Annual median FT-IR spectra corroborated the trends in FG concentrations, as did monthly and seasonal observations. Monthly median OM concentrations were generally higher in the warmer months, in line with predominant biogenic secondary OM sources; this was additionally





corroborated by the greater COOH contribution to OM in the summers. A greater aCH contribution to OM was observed in the winters, as were higher urban/rural ratios of OM, suggesting more anthropogenic carbon emissions during the cold season. The two observed FG trends were suggested to originate from separate phenomena. Declines in the more oxygenated FGs were attributed to declining secondary OM from anthropogenic VOC emissions and/or $SO_2$ concentrations; the latter

directly affects $SO_4^{2-}$ concentrations and implies one of a handful of indirect mechanisms suggested elsewhere (Blanchard et al., 2016; Marais et al., 2017; Xu et al., 2015b). The hypothesized sources of the more oxygenated FG declines were supported by several observations: (a) summertime SE U.S. OM measured in other studies had predominant fractions from $SO_4^{2-}$-associated OC and isoprene (Blanchard et al., 2016; Xu et al., 2015b); (b) COOH was prominent in mixed oxidized OM and fossil fuel combustion OM in prior FT-IR spectrometry studies (Sect. 3.3.1); (c) the maximal declines in OM and

oxidized FG (oxOCO and COOH) concentrations were in summertime, suggesting an isoprene carbon source and/or a photochemical formation mechanism; and (d) the decline spatially matches that of $SO_4^{2-}$, another secondary product of the atmosphere. While there are multiple plausible hypotheses for the source of the aCOH and naCO fractions of OM, which trended separately from the more oxygenated FGs: the enhancement in low-$NO_x$ secondary OM from monoterpene emissions is most likely, based on prior factor-based FT-IR spectrometry literature on these two FGs. Increasing aCOH and

naCO concentrations corresponded seasonally with monoterpene emission events and enhancements in $O_3$ concentrations, suggesting either a link between $NO_x$ and $O_3$ in this pathway, or that $O_3$ may be independently involved in the increase in FG concentrations.

OM/OC estimated from functional group measurements did not significantly change over time due to the competing enhancement/decline of the oxygenated FGs. OM/OC values were greater in summers and at rural sites, as anticipated, and

the magnitudes were in agreement with previous work using similar methodology (Bürki et al., 2020). In contrast, the trend in OM/OC ratios previously estimated by incorporating TOR OC measurements and other techniques (Hand et al., 2019) differed from that observed in the current study. Additional comparisons of OM/OC measurements should be made.

Events such as fires and stagnation, which have been suggested by chemical tracers and meteorological data elsewhere (Hidy et al., 2014; Zhang et al., 2010; Zheng et al., 2002), can be visualized in the daily OM and FG concentrations. While trends

in absolute OM concentrations were often dominated by anthropogenic and/or biogenic secondary OM, variability in the shorter-term (as demonstrated by daily data in 2016) was often related to smoke and precipitation events. During the winter/spring (January through April), smoke from prescribed burning activities appears to have contributed many of the higher OM concentration samples. Summertime (April through August) OM concentrations showed periods of build-up due to secondary biogenic OM followed by rainout. Severe fires in November generated the greatest 2016 daily OM

concentrations, but elevated aCH contributions to the OM persisted throughout November and December, perhaps suggesting residential wood burning emissions (which are a substantial source of OM in the SE U.S.; Chen et al., 2012; Sun et al., 2011b).



This work additionally served to evaluate the applicability of FT-IR spectrometry and PLS-based FG models for routine OM characterization, for air quality monitoring networks or extended studies. As also validated in part one of this set of papers (Boris et al., 2019), the measurements demonstrate general similarity in magnitude to other OM characterization studies.

This work adds to the growing literature on SE U.S. aerosol formation, including its evolution as anthropogenic emissions

decrease. These FT-IR spectrometry and PLS analysis results corroborate the presence of multiple fractions of OM as observed in other work (Liu et al., 2018; Xu et al., 2015b), but highlight the different trajectories of those fractions within the 2011–2016 time frame. The agreement between FG data and other work supports the application of FT-IR and PLS methodology to routine (multi-annual) studies, which has not been investigated before. Results demonstrated, in particular, the success of pairing the concise information provided by FGs with shorter-term studies that used more intensive techniques

(such as aerosol mass spectrometry or organic molecular tracer analyses). The daily patterns observed in the present work also demonstrate the utility of characterizing FG OM to easily elucidate shorter-term carbonaceous aerosol dynamics, including the impacts of fires and meteorological events.

*Data availability.* The functional group, OM, and OC concentrations with uncertainties, as well as raw spectra, are available
at https://doi.org/10.25338/B8SG73 (Dillner et al, 2019).

*Author contributions.* AMD and SLS conceived of the project. AMD provided leadership for the project including mentoring and supervising AJB in data analysis, interpretation, and final review prior to publication. AJB performed the data analysis and interpretation, created the figures, and wrote and edited the manuscript. ST provided additional mentoring to AJB on
data analysis and interpretation and editing of the manuscript. ATW and BMD provided input on the data analysis and reviewed the paper. ESE and SLS provided SEARCH network filters and data. TJ and NLN provided additional samples for the Atlanta site after the SEARCH network was shut down and reviewed the paper.

*Competing interests.* The authors declare that they have no conflict of interest.

*Acknowledgements.* Funding for this project was generously provided by the Electric Power Research Institute, with equipment and logistical support from Atmospheric Research & Analysis, Inc. T. J. and N. L. N. acknowledge support from CAREER AGS-1555034. Special thanks to Kelsey Seibert, who skilfully managed the collection of FT-IR spectra and sample handling. Thank you to Matthew Coates, Martin Esparza–Sanchez, Carley Fredrickson, Nathaniel Hopper, and Alex
Williams, who worked to collect calibration standards and perform various essential data analysis functions. The authors also acknowledge Jenny Hand for her guidance on OM/OC calculation, Charlotte Burki for her work in calculating the OM/OC ratios via regression methods, and Sean Raffuse for his support with fire detection methods and data.

*Financial support.* This research has been supported by the Electric Power Research Institute (grant no. 10003745).





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
