# Peer review of "Quantifying organic matter and functional groups in particulate matter filter samples from the southeastern United States – Part 2: Spatiotemporal Trends"

_Atmospheric Measurement Techniques, 2020_

## Author Response (AR1)

**Response to Reviewers: for "Quantifying organic matter and functional groups in particulate matter filter samples from the southeastern United States, part 2: Spatiotemporal Trends" by A. J. Boris et al. (amt-2020-401)**

**Interactive comment from Anonymous Referee #1**

*Please see responses in italics*

This manuscript reports an original analysis of 6 years of filters collected from 4 different monitoring sites in the SE US. The application of the method is novel and the analysis is original. The results provide interesting insights on trends in PM2.5 during this period, which could fit equally well in ACP rather than AMT. Authors provide substantive comparisons of their results to relevant literature, and generally this is well done and sufficient for publication. In sum, this is a very high-quality set of original measurements by a very experienced group and merits publication.

*We appreciate your positive assessment of our work.*

My only misgiving is the extent to which the technique is different from the cited work. I ask the authors to revise the manuscript to be specific about this to merit publication in AMT rather than ACP.

*We have adjusted our manuscript to emphasize the differences between our work and other cited work in the following ways:*

- *Added to the abstract: "To the best of our knowledge, this is the longest time period over which this type of analysis has been applied, and this work also demonstrates the application of a more chemically complete and less destructive method than in prior work using alternate techniques."*
- *In Section 1.3, the "Research Statement", we added a clause to distinguish the role of our first paper within this series: "FG calibration models described by Boris et al. (2019; part 1 of this study, which describes the method development) were applied to SEARCH spectra in the current study to accomplish the following goals"*
- *In Section 1.3, the "Research Statement", added a sentence to highlight the novel measurements being made in this paper: "We herein present the application of FT-IR spectrometry to eight years of routinely collected SEARCH network PTFE ambient aerosol filter samples to examine trends in OM concentrations, OM/OC and their FG composition from 2009 to 2016. These quantities have not been directly measured in SEARCH network data prior to this work."*
- *In the Conclusions, added a sentence at the beginning to summarize the distinct work discussed in this manuscript: "Multi-annual, seasonal, and daily trends in OM concentrations and composition were examined in $PM_{2.5}$ from the SE U.S. using FT-IR spectrometry and PLS regression."*

*We thank the reviewer for suggesting that our work might merit publication in ACP; however, we would like to publish in AMT as this paper would the second in a series, and we selected this journal for this pair of papers recognizing that neither AMT nor ACP would be a perfect fit for both works.*

As a note, I find the most interesting part of the discussion to be about the contribution of fires to particular functional groups, but that seems a bit buried in the Discussion and is omitted from the Abstract, Title, and Conclusions.

*We agree that the apparent correspondence of observed fires and variability in the daily (2016) OM concentrations is an interesting observation. Unfortunately, a smoke "fingerprint" within FG composition was not possible to discern in this work: as demonstrated in figure S1, while one composition of fire-impacted samples (as identified using satellite images) was observed in daily 2016 samples, another was observed in 2011-2016 samples collected every three days. We believe that this satellite imagery-based method of categorizing "fire" vs. "no fire" samples, as well as other factors affecting smoke composition and dilution arriving at the sampling sites, were responsible for this lack of observable smoke FG "fingerprint". We have updated our text to be clearer about this in the manuscript (Section 3.6, page 20).*

*We have also included clarification on the discussion of fire contributions in the following ways:*

- *Added the ending clause to this statement in the abstract: "Daily samples from 2016 further elucidate the consistent impact of meteorology and biomass burning events on shorter term OM variability, including prescribed burning in the winter/spring and wildfires in the autumn, although these sources did not appear to be strong contributors to long-term OM or composition trends in the SE U.S."*
- *Added this sentence in the conclusions: "A fire "fingerprint" of FG composition was not, however, apparent in the present work, despite the clear relationship with overall OM concentrations, perhaps due to dilution or other obfuscating factors."*

To that point the authors note the potential usefulness of PMF, but apparently did not have the time/resources to do this, which seems a shame. Perhaps future work on this is planned?

*Unfortunately, PMF was outside the scope of this particular paper, but we agree that it could be quite useful, and hope to perform PMF analysis using these data and similar datasets in the future.*

Other questions:

1. to volume or are volumes same?

*All samples in this work were collected over 24 hours at 16.7 liters per minute (24.0 $m^3$). Section 2.5 (page 7) states that, "when available, if the observed flow rate, duration, or total volume was not 100±5 % of the expected value (e.g., <23.75 of 24.0 expected hours), the sample was excluded from the dataset; samples with three or more null flow observations were assumed to be inaccurately sampled and were also excluded."*

*The following text was added to Section 2.1 (page 5) to clarify this: "All samples were collected over 24 hours at 16.7 liters per minute (24.0 $m^3$; see Section **Error! Reference source not found.** for outlier handling). Filters were shipped and stored at < 4 ℃ (from Aerosol Research and Analysis, Inc., ARA, in Morrisville, NC) to minimize loss of volatile species."*

2. Fig4 annual median spectrum is an interesting concept; can you provide more details about how this was determined?

*We kindly refer the reviewer to our Supplement, Section S5, which contains a description of the method of median spectrum calculation (briefly, baseline correction and blank filter feature subtraction followed by median calculation at each wavelength). Figure S6 also shows the median spectra of other sites and years.*

3. Table 2 – trends are not consistent; can you provide explanations for lack of consistency?

*The trends in this table are expressed in terms of the changes in functional group (FG) concentrations over time, and the changes in the ratios of FG concentration to the total organic matter (OM) concentrations (FG/OM; Figure 3 trends are also as FG/OM). The normalization to OM in the latter category causes some trends to differ substantially in magnitude, and even direction, in some cases.*

*In response to this and another related reviewer comment, we have moved this table to the Supplement. Important trend values have been added within the text of the manuscript. We hope that the manuscript is more succinct without this complex table.*

**Interactive comment from Anonymous Referee #2**

This manuscript describes FTIR analysis coupled with multivariate calibration of specific functional groups (aliphatic C-H, carboxylic COOH, oxalate O=C-O-, non-acid and carbonyl C=O and alcohol O-H) in filter-based fine PM samples collected at the surface over 8 years at 4 sites in the southeastern United States as part of the former SEARCH air quality network. The authors find that there is a decline in organic mass that is driven primarily by carboxylic acid and oxalate functional groups. They attribute this to reductions in anthropogenic emissions of SO2 and/or VOCs. There is a lot of analysis and the supplemental information is extensive.

*We appreciate your positive assessment of our work.*

Control experiments and quality assurance are discussed in the supplemental information (SI). It is difficult to understand the impact of not storing samples frozen over several years, and the SI indicates there is chemical change. Looking at Figure 4, my first instinct was the annual trend is related to sample degradation - I actually think a more thorough discussion in the main text would help the authors' case.

*This critique provides an important perspective for us on this matter. We have moved material from the SI to the main text to more visibly discuss the possibility of sample degradation as a cause of OM decline over the sample years.*

*In order to further clarify and support our hypotheses, we have taken the following measures:*

- *Discussion points from our first paper, where these storage studies were initially reported, have been moved to the Discussion section of this paper (Section 3.2; page 13-14). Specifically, we have expanded upon the variability of re-analyzed filters and differences between shorter- and longer-term storage. We have additionally highlighted changes in COOH and oxOCO in particular, since these two FGs were observed to decline over time in the present work.*

- *The observation that the 2009-2010 FG and OM concentrations were actually increasing (due to the 2008 recession) also supports the lack of OM loss during storage; this has been further highlighted in Section 3.2 (page 13-14).*

- *The cold storage of filters (< 4 ∘C) to minimize loss of volatile species is now mentioned in the Methods section (Section 2.1, page 5) and within the Discussion section (Section 3.2, pages 13-14).*

Further, the authors attempt to make links to temporal trends among $NO_x$ or $O_3$ with specific FTIR-derived functional groups almost exclusively with literature concerning ambient data. It's my opinion that links to laboratory literature with reference to specific spectra wavenumbers help make their arguments stronger.

*There have only been limited studies of laboratory (chamber) experiments published thus far using infrared spectrometry, and studies quantifying specific molecules (for which FG composition is known) explain only a small fraction of the overall OM mass. Due to aging times, specific fuels or precursor molecules, and photooxidation conditions, the resonances of functional groups in condensed molecules may vary according to a variety of intramolecular and*

*intermolecular interactions. Therefore, broad ranges rather than specific wavenumbers are used to interpret the FG composition in this study. However, biogenic secondary organic aerosols observed in various ambient environments are consistent with reported FG composition and spectral profiles due to oxidation of biogenic precursors in chamber experiments (for example, Corrigan et al., 2013; Liu et al., 2018).*

*Corrigan, A. L., Russell, L. M., Takahama, S., Aijälä, M., Ehn, M., Junninen, H., Rinne, J., Petäjä,T., Kulmala, M., Vogel, A. L., Hoffmann, T., Ebben, C. J., Geiger, F. M., Chhabra, P., Seinfeld, J. H., Worsnop, D. R., Song, W., Auld, J., and Williams, J.: Biogenic and biomass burning organic aerosol in a boreal forest at Hyytiälä, Finland, during HUMPPA-COPEC 2010, Atmospheric Chemistry and Physics, 13, 12 233–12 256, doi:10.5194/acp-13-12233-2013, 2013.*

*Liu, J., Russell, L. M., Ruggeri, G., Takahama, S., Claflin, M. S., Ziemann, P. J., Pye, H. O. T., Murphy, B. N., Xu, L., Ng, N. L., McKinney, K. A., Budisulistiorini, S. H., Bertram, T. H., Nenes, A., and Surratt, J. D.: Regional Similarities and NO$_x$-Related Increases in Biogenic Secondary Organic Aerosol in Summertime Southeastern United States, Journal of Geophysical Research: Atmospheres, 123, 10,620–10,636, doi:10.1029/2018JD028491, 2018.*

*In response to the reviewer's concern, we have simplified and moved much of the material in this literature-based discussion to the Supplement. The discussion in the manuscript highlights the findings of other FT-IR spectrometry studies, including the work cited above, and particularly work done using samples from the SE U.S. However, as mentioned above, the FT-IR spectrometry literature is still somewhat limited, so we hope to only suggest probable links to atmospheric sources or processes in other discussions.*

I found Table 2 very confusing. It appears the aCH trend is driven by 'just' BHM summer, COOH 'just' in the summer, but many other values are presented. I understand how this would be value for careful readers. However there are more compelling plots in the SI that I think would make better use of the main text space.

*We appreciate the reviewer's precise suggestions for improving the clarity of our paper. We have moved Table 2 from the main text to the Supplement and have embedded important values from that table within the text. In an effort to keep our manuscript succinct, we have decided not to move additional figures from the Supplement.*

Page 4, Line 16: The authors state fossil fuel combustion may contribute substantially to OM in the SE U.S. …. I think the authors here are referring to the carbon component specifically. It is well established that fossil fuel combustion aids OM formation, e.g., impacts on the NO$_x$/oxidants and POA are known for over a decade...work by Lane, Griffin, Carlton.

*Upon considering the literature suggested by the reviewer, we have altered the text here to specify known effects of fossil fuel combustion on OC and OM concentrations. The text now reads (Section 1.2, page 4):*

"*Fossil fuel combustion contributes substantially to OC in the SE U.S., especially at urban sites: ~50 % of primary and secondary OC at urban BHM was estimated to be from fossil sources, but at rural CTR, >80 % of primary and all of secondary OC was from modern sources such as trees (in the early 2000s; Blanchard et al., 2008). In addition, other emissions from fossil fuel combustion, including NO$_x$, SO$_2$, and primary OM, also affect OM concentrations by modulating*

*aerosol surface area, aerosol water content, aerosol acidity, and oxidant concentrations (Nguyen et al., 2015; Carlton et al., 2010; Al-Naiema et al., 2018)."*

*Al-Naiema, I. M., Hettiyadura, A. P. S., Wallace, H. W., Sanchez, N. P., Madler, C. J., Cevik, B. K., Bui, A. A. T., Kettler, J., Griffin, R. J., and Stone, E. A.: Source apportionment of fine particulate matter in Houston, Texas: insights to secondary organic aerosols, Atmos. Chem. Phys., 18, 15601–15622, https://doi.org/10.5194/acp-18-15601-2018, 2018.*

*Carlton, A. G., Pinder, R. W., Bhave, P. V., and Pouliot, G. A.: To What Extent Can Biogenic SOA be Controlled?, Environ. Sci. Technol., 44, 3376–3380, https://doi.org/10.1021/es903506b, 2010.*

*Nguyen, T. K. V., Capps, S. L., and Carlton, A. G.: Decreasing Aerosol Water Is Consistent with OC Trends in the Southeast U.S., Environ. Sci. Technol., 49, 7843–7850, https://doi.org/10.1021/acs.est.5b00828, 2015.*

Page 11, Line 6/7: I do not follow the logic behind the statement that OM trends are unrelated to changing PM$_{2.5}$ concentrations due to lack of trend in Si and K. These species

*We have removed this sentence from the manuscript; it was unnecessary.*